# Streaming Radiance Fields for 3D Video Synthesis

**Lingzhi Li**
Alibaba Group
llz273714@alibaba-inc.com

**Zhen Shen**
Alibaba Group
zackary.sz@alibaba-inc.com

**Zhongshu Wang**
Alibaba Group
zhongshu.wzs@alibaba-inc.com

**Li Shen**
Alibaba Group
lshen.lsh@gmail.com

**Ping Tan**
Alibaba Group
pingtan@sfu.ca

## Abstract

We present an explicit-grid based method for efficiently reconstructing streaming radiance fields for novel view synthesis of real world dynamic scenes. Instead of training a single model that combines all the frames, we formulate the dynamic modeling problem with an incremental learning paradigm in which per-frame model difference is trained to complement the adaption of a base model on the current frame. By exploiting the simple yet effective tuning strategy with narrow bands, the proposed method realizes a feasible framework for handling video sequences on-the-fly with high training efficiency. The storage overhead induced by using explicit grid representations can be significantly reduced through the use of model difference based compression. We also introduce an efficient strategy to further accelerate model optimization for each frame. Experiments on challenging video sequences demonstrate that our approach is capable of achieving a training speed of 15 seconds per-frame with competitive rendering quality, which attains $1000\times$ speedup over the state-of-the-art implicit methods.

## 1 Introduction

3D video synthesis aims to realize free-viewpoint photo-realistic rendering for dynamic scenes, which are typically recorded via a set of cameras (with known poses) from multiple views. The topic has attracted much research effort because of its potential value in a wide range of applications in VR/AR. Traditional techniques by estimating surfaces [1], multi-sphere images [2] or depth [3, 4] via multi-view stereo [5, 6] are usually used for modeling and representing dynamic scenes. However, arbitrary geometry and complex appearance exhibiting in real-world scenarios pose challenges for leveraging a general methodology to pursue high-performance modeling.

Neural radiance fields (NeRF) [7, 8] have recently emerged as a new methodology for effectively reconstructing and rendering static scenes via neural networks. These methods learn a continuous mapping between 3D points (given view directions) and their corresponding radiance colors and opacity, to realize high-fidelity rendering results through the use of volumetric rendering techniques [9]. However, they often suffer from costly training and inference due to a tremendous amount of computations through neural networks. When extending the implicit formulation to a dynamic scene [10], time steps are embedded as additional input for training models across frames and achieving time-dependent rendering. Compared to training on static scenes, training overhead increases significantly with respect to sequence length, e.g., costing about 56 GPU days for training 300 frames, that would be prohibitive when handling long sequences. More importantly, such learning paradigm is restricted to offline modeling, thus unable to tackle online scenarios.

36th Conference on Neural Information Processing Systems (NeurIPS 2022).

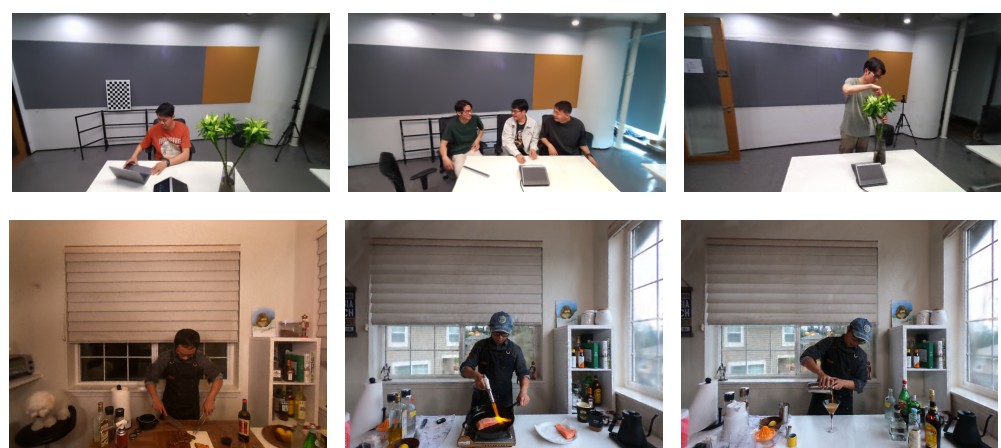

Figure 1: **Rendering results on test view.** *Top*: Our meet room dataset; *Bottom*: N3DV dataset. These novel view results are rendered at interactive speed ($\sim$10 FPS) by StreamRF.

In this work, we propose a novel method for reconstructing streaming radiance fields for novel view synthesis of 3D videos. We formalize the modeling of a dynamic scene in an incremental learning framework, enabling it to tackle video sequences on-the-fly. Inspired by the recent success achieved by Plenoxel [11] on training and inference efficiency, our method is built on explicit grid representations. Specifically, the overall training process is decomposed to the complete training of a basic voxel grid model at the first frame, with an online tuning mechanism for learning and storing model difference for each subsequent frame. A complete grid model given a time step $i$ can be simply attained by adding the model difference to the previous model at $i-1$ (or accumulating all the model differences before $i$ with the basic grid model). By exploiting the prior of exhibiting local continuity between adjacent frames, we present a narrow-band tuning strategy for capturing and training the model difference efficiently, associated with a compression strategy for significantly reducing memory cost that explicit grid methods struggle with. To further improve the training efficiency, we propose to learning pilot models to guide the optimization of full-scale models. The overall framework is simple yet efficient. Quantitative and qualitative experimental results demonstrate the effectiveness of our approach. The approach can achieve the training speed of 15 seconds and inference time of 120 ms per frame for rendering a 1k resolution image with competitive quality, achieving three orders of magnitude faster than the state-of-the-art dynamic neural radiance field methods (costing 56 GPU-days for training a sequence with 300 frames). Compared to the efficient explicit-grid baseline by per-frame training, it still can obtain ~100× acceleration with a much lower storage cost. The empirical analysis further demonstrates that using a proper incremental training mechanism would not degrade performance when dealing with relatively long sequences. Code is available at https://github.com/AlgoHunt/StreamRF.

## 2  Problem Formulation

Neural Radiance fields (NeRF) [7] model a *static* scene implicitly by learning a continuous mapping from a 3D spatial position $\mathbf{x} \in \mathbb{R}^3$ specified in a viewing direction $\mathbf{d} \in \mathbb{R}^3$ (represented as Cartesian unit vectors) to RGB radiance color $\mathbf{c} \in \mathbb{R}^3$ and opacity $\sigma \in \mathbb{R}$. The mapping function is typically implemented by a multilayer perceptron (MLP) $f_m : (\mathbf{x}, \mathbf{d}) \mapsto (\mathbf{c}, \sigma)$, trained on a set of images with known camera pose. Given a ray $\mathbf{r}$ casting on the image plane through a pixel, by using a traditional volume rendering technique [12], the expected color $\widehat{C}(\mathbf{r})$ of the pixel is rendered by integrating the opacity and the color of sampling points along the ray:

$$\widehat{C}(\mathbf{r}) = \sum_{i=0}^{N-1} T_i \left(1 - \exp(-\sigma_i \delta_i)\right) \mathbf{c}_i, \quad T_i = \exp\left(-\sum_{j=0}^{i-1} \sigma_j \delta_j\right), \tag{1}$$

where $\delta_i$ denotes the distance to adjacent points. These methods are subjected to slow training and rendering speed, since the ray casting involved in both training and rendering stage requires hundreds

of forward propagation (and backward propagation in training) through MLPs for the sampling points on each ray. For example, JaxNeRF [13], though optimized through parallelization, needs over 20 seconds to render an image with $800 \times 800$ resolution with over 10 hours of training.

When extending the mechanism to model a *dynamic* scene, one straightforward solution is to embed the time variable $t$ into the representation, i.e., $f_t : (\mathbf{x}, \mathbf{d}, t) \mapsto (\mathbf{c}, \sigma)$, while training time would dramatically increase when dealing with a large number of frames. Training a video with original NeRF frameworks may cost over 600 GPU-days. N3DV [10] improves training efficiency by virtue of sophistical designs, however it still needs about 56 GPU-days for pursuing a model with reliable rendering quality. More importantly, these implicit formulations are naturally suitable for offline processing which feeds all the (key)frames in advance for training, and may not be easily generalized to online training scenarios.

Recent progress on explicit representations has demonstrated compelling improvement on training and render efficiency for modelling a *static* scene. The scene space is discretized and represented explicitly with a sparse voxel grid instead of learning representations implicitly with large MLPs. As a representative work, Plenoxel [11] optimizes voxel grids without neural networks. Each voxel in the grid consists of the scalar opacity $\sigma$ and the coefficient vector $\mathbf{k}$ of spherical harmonics (SH) for respectively describing geometric and color information. The color at the voxel $v$ is obtained by,

$$\mathbf{v}_c = c(\mathbf{d}, \mathbf{k}) = \xi \left( \sum_{l=0}^{l_{max}} \sum_{m=-l}^{l} \mathbf{k}_l^m \cdot Y_l^m(\mathbf{d}) \right), \tag{2}$$

where $\xi$ is the sigmoid function and $\{Y_l^m\}$ denotes the set of SH basis. To render the expected color of the pixel through a ray, the opacity and the color of a sampling point $\mathbf{x}$ are calculated by performing trilinear interpolation among the voxels within its neighborhood $\mathcal{N}(\mathbf{x})$,

$$\sigma = U_{\mathbf{v} \in \mathcal{N}(\mathbf{x})} (\mathbf{v}_\sigma), \quad \mathbf{c} = U_{\mathbf{v} \in \mathcal{N}(\mathbf{x})} (\mathbf{v}_c). \tag{3}$$

Then the pixel color is produced by integrating the sampling points along the ray based on Eq. 1. The voxels with zero opacity (or truncated by a minimal threshold) are pruned after sparsification, and an occupancy mask $\mathbf{M}$ with binary values is used for indicating if the space is occupied. More details can refer to the paper of Plenoxels [11]. In theory, the mechanism could significantly reduce the overall computational complexity as it omits the requirement of numerous MLP-based costly computation during training and inference. It is able to train a static scene with good rendering quality within 15 minutes on single GPU and render a $1k$ resolution image in $100$ ms. These efficiency gains would benefit developing a highly efficient framework for modeling and rendering a dynamic scene.

## 3 Streaming Radiance Fields for Dynamic Scenes

To realize sparse grids for the modeling of a dynamic scene, one intuitive way is to directly extend 3D spatial dimensions (denoted by $H \times W \times D$) with the 4-th dimension indicating the time steps $t$ for frames. However, using explicit grids for a static scene have faced challenges on storage although progressive training and sparsifying (i.e, pruning off empty voxels) [11] are used to alleviate the issue of costly overhead. Simply expanding the grid into a space-time manner would make storage overhead increase linearly with $t$, which is infeasible for handling videos with long sequences. On the other hand, per-frame training individually without considering temporal correlation may be feasible for achieving a tradeoff between storage and efficiency, while it is still far from the objective of pursing a high-performing framework for 3D video synthesis.

In this regard, we propose a novel method, named StreamRF, to realize a streaming radiance field for effectively rendering dynamic scenes. Our approach is built on the benefits of utilizing an explicit representation via sparse voxel grids. Particularly, by leveraging some intrinsic priors exhibiting in videos, we develop an incremental training framework which enables a faster training convergence and lower storage over per-frame grid modelling.

Videos in forward-facing scenarios are typically composed of time-invariant components (e.g., static background) and time-variant components (e.g., moving objects) with small variation that exhibits local continuity between adjacent frames. By exploiting the fact, we decompose the problem of sequential modeling into two sub-problems. We first fully train a base grid model given the first frame, denoted as $\mathbf{V}^0 = \{\mathbf{V}_c^0, \mathbf{V}_\sigma^0\}$ which stores the corresponding color features (the coefficients of

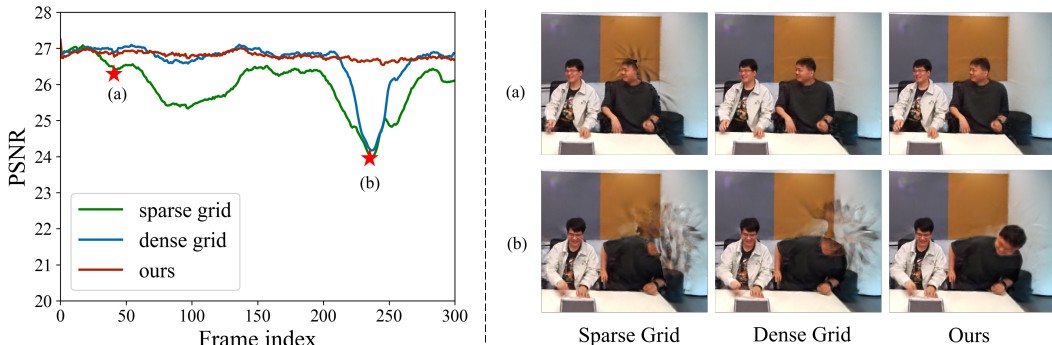

Figure 2: *Left*: PSNR comparison between tuning with (1) original sparse grid (2) dense grid and (3) our narrow band finetune in a 300 frames sequence; *Right*: visual comparison of above methods on (a) $40th$ frame (b) $240th$ frame. All results are trained with the same initial model.

spherical harmonics) and opacity values. Then the model is adapted to succeeding frames through the use of an effective tuning strategy. Per-frame tuning is expected to perform only on the changes between adjacent frames. The grid model in the current frame is produced by adding the model difference, which is optimized and stored during training phase, to the grid model in the previous frame. Formally, assume the model difference between the $i-1$-th and $i$-th frames is denoted as $\delta_i = \Delta(\mathbf{V}^i, \mathbf{V}^{i-1})$. In the inference phase, the grid model at the time step $i$ can be obtained by,

$$\mathbf{V}^i = \mathbf{V}^{i-1} + \delta_i = \mathbf{V}^0 + \sum_{j=1}^{i} \delta_j. \tag{4}$$

Such an incremental formulation indeed presents an online training paradigm, enabling the proposed framework to handle video sequences on-the-fly.

### 3.1 Tuning with Narrow Bands

A problem arises: how to effectively capture the model difference between adjacent frames $i-1$ and $i$. Given the sparse grid model $\mathbf{V}_{i-1}$, any object motion at the time step $i$ may result in changes on the opacity and colors of a part of voxels. It is hence likely to happen that informative points occur in the originally empty space as well as some voxels originally occupied becomes empty. If we directly tune the sparse grid, only the non-empty voxels could be changed or pruned in a one-way manner while unable to add informative voxels. It is problematic when handling long sequences (shown by the ablation study on the sparse grid in Fig. 2). To address the issue, we propose a tuning strategy based on narrow bands. We first retrieve and activate the narrow band for the frame $i$, i.e., the region of interest in the grid $\mathbf{R}^i$, based on the occupancy mask $\mathbf{M}^{i-1}$ (with binary values) which indicates non-empty voxels at the frame $i-1$,

$$\mathbf{R}^i = \mathcal{F}_{di}(\mathbf{M}^{i-1}, \rho_d) \oplus \mathcal{F}_{er}(\mathbf{M}^{i-1}, \rho_e), \tag{5}$$

where $\mathcal{F}_{di}$ and $\mathcal{F}_{er}$ correspond to dilation and erosion operations, and $\rho_d$ and $\rho_e$ denote the corresponding radius parameters. $\oplus$ denotes the element-wise xor operation on the masks to extract the additional region of the dilated area compared to the eroded one. The narrow band actually represents a thin area around the surface of scenes (or objects in a composite scene). We use it for two purposes:

- For every empty voxel that falls into the band, we restore them to the grid and initialize the opacity and color features with zero values.

- Only the voxels within the band are tuned and the rest ones are not involved during the training stage for the frame $i$.

In other words, the tuning always performs on a relative small set of voxels in the frame $i$. We also prune empty voxels off to get a compact grid $\mathbf{V}_i$ with an occupancy mask $\mathbf{M}_i$ after tuning.

The key idea of narrow band-based tuning is built upon leveraging the intrinsic priors: motions induced by moving objects typically show small changes and local continuity within neighboring

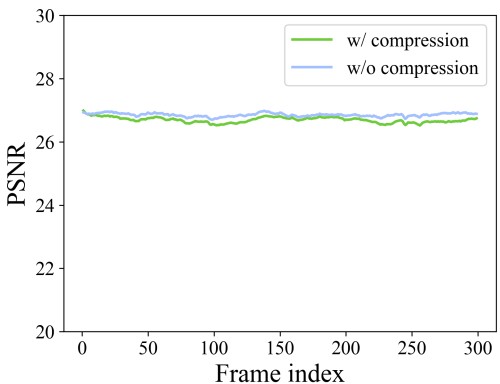 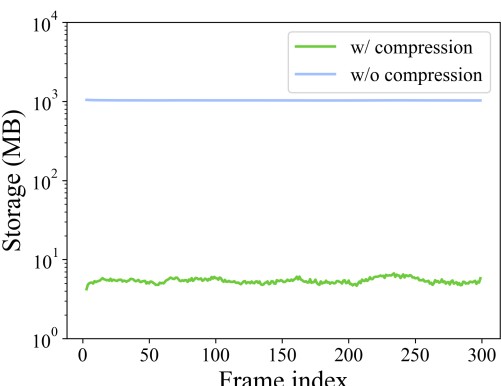

Figure 3: **Ablation study of diff-based compression**. We compare per frame PSNR (**left**) and storage size (**right**) with diff-based compression enabled and disabled. The storage space drops to 0.5% of the original with a negligible difference on PSNR (decrease 0.156 in average)

areas between adjacent frames. These assumptions have been successfully exploited in a wide range of video processing [14] and video understanding tasks [15, 16], e.g, traditional mean-shift method for tracking [17]. We demonstrate that the strategy can effectively benefit to training convergence and be capable of achieving the speed of $\sim 15$s per-frame training for achieving a competitive quality (shown in section 5.4). Compared to the state-of-the-art implicit method [10] with the training speed of 4.5 GPU-hours for each frame in average, our method is about three orders of magnitude faster on training and can achieve over $500\times$ speedup on rendering efficiency (over 60 s s VS 121 ms).

## 3.2 Difference-based Compression

Using explicit grid modelling may pose the issue of large storage cost. Taking a grid size ($1408 \times 1156 \times 128$ in the experimental setting) for example, the overall storage for a conventional sparse grid costs about 1015 MB after sparsity for one frame. By virtue of narrow band tuning, we only need to store the model difference $\delta_i$ with a significantly smaller size compared to the complete grid model. The change of storage cost only happens in the following two scenarios, i.e., when adding voxels into the grid (i.e., appearing in the empty space) and pruning voxels off from the grid (i.e., the previously occupied space becomes empty). In this regard, we introduce a difference-based compression strategy to further reduce the storage overhead. We can efficiently find out the change areas by using the occupancy masks $\mathbf{M}^{i-1}$ and $\mathbf{M}^i$ between the adjacent frames $i-1$ and $i$,

$$
\begin{aligned}
\mathbf{M}_a^i &= \mathbf{M}^i \odot (1 - \mathbf{M}^{i-1}), \\
\mathbf{M}_e^i &= \mathbf{M}^{i-1} \odot (1 - \mathbf{M}^i), \\
\mathbf{M}_r^i &= \mathbf{M}^{i-1} \odot \mathbf{M}^i \odot \mathbf{1}\{\gamma(\delta_i^c) > \epsilon\},
\end{aligned}
\tag{6}
$$

where $\mathbf{M}_a^i$, $\mathbf{M}_e^i$ and $\mathbf{M}_r^i$ respectively denote the masks of adding voxels, erasing voxels and remaining ones while the features at the voxels may change. $\gamma(\cdot)$ defines the function of getting the L1 norm along the axis of color features (SH coefficients) on the model difference, and $\epsilon$ denotes a threshold. $\mathbf{M}_r^i$ actually involves the voxels with relatively large modification. Instead of storing model difference $\delta_i$ directly, we store the three masks, the voxel representations (SH coefficients and opacity) belonging to the adding area $\mathbf{M}_a^i$ and the representation differences for the voxels within the remaining area $\mathbf{M}_r^i$. It can achieve a remarkable reduction rate (178 times) on storage that reduces **1015 MB** to **5.7 MB** for each frame in average.

## 4 Efficient Training with Pilot Model Guidance

To further improve the training efficiency of StreamRF, we exploit an optimization strategy inspired from curriculum learning to train from an easier problem. Formally, given the grid model $\mathbf{V}^{i-1}$, we adopt down-sampling operations to get a *pilot* model of a smaller size,

$$
\widehat{\mathbf{V}}^{i-1} = \downarrow^S (\mathbf{V}^{i-1}).
\tag{7}
$$

When setting the $S$ to $1/2$, the capacity of pilot model is roughly $1/8$ of the full grid. Training on a smaller model would reduce the computational complexity at every iteration (with the same batch size) and also ease optimization. The model difference on the pilot models $\widehat{\delta}_i = \Delta(\widehat{\mathbf{V}}^i, \widehat{\mathbf{V}}^{i-1})$ are implemented by using $\{\widehat{\mathbf{M}}_a^i, \widehat{\mathbf{M}}_e^i, \widehat{\mathbf{M}}_r^i\}$ as following Eq. 6. Then we can get a guidance mask $\widehat{\mathbf{G}}^i$ by merging the three mask via elementwise "or" operator and upsampling to the full scale,

$$\widehat{\mathbf{G}}^i = \widehat{\mathbf{M}}_a^i \vee \widehat{\mathbf{M}}_e^i \vee \widehat{\mathbf{M}}_r^i,$$
$$\mathbf{G}^i = \uparrow^S (\widehat{\mathbf{G}}^i). \tag{8}$$

The guidance mask is used for regularizing the gradient update when tuning the full-scale model. The values of added and erased voxels are filled back to the full-scale grid. All the voxels outside the mask are frozen without optimization when training at the frame $i$.

The design of the strategy comes from the insight that training at a small size converges much faster and the induced difference of pilot model indicates the voxels that need to be modified. Using this strategy can prevent unnecessary modification in the full-scale model and avoid the loss of high-frequency details during downsample-upsample process. We observe that using this strategy facilitates pursuing a higher fidelity model compared to directly training on the full-scale grid with the same training time. The qualitative results (in Fig. 4) reveals that the model via directly training on the full-scale grid may encounter flicker issues when rendering along time steps, i.e., sparkling noise may occur in the stable background area when training period is short. The issue could be addressed by lengthening the training schedule with a small learning rate. While training with the pilot model guidance can effectively prohibit flicker artifacts from happening, enabling a more efficient and stable optimization for the streaming paradigm.

# 5 Experiments

## 5.1 Datasets

**Meet Room Dataset.** We build a multi-view capture system using 13 Azure Kinect cameras. We use twelve cameras for training and one for testing, with an external pulse signal to synchronize all camera shutter. Though this system is capable of depth capture, we do not use depth images in this research. The videos are recorded at a resolution of $1280 \times 720$ and a frame rate of 30 FPS. The total size of the capture system is 100 cm $\times$ 75 cm and the captured views are aligned much sparser than some common datasets such as LLFF [18] and N3DV [10]. We will release this dataset for research purposes.

**Neural 3D Video (N3DV) Dataset [10].** It contains 6 dynamic indoor scenes with varying illuminations, view-dependent and highly volumetric effects. Videos are captured at a resolution of $2704 \times 2078$ and a frame rate of 30 FPS. Following the setting in the original paper [10], the quantitative score is produced on the designated sequence ("flame_salmon") by downsampling it to $1352 \times 1039$ for a fair comparison, where the frames captured by 18 views are used for training and the rest one for testing.

## 5.2 Implementation Details

**Initialization.** We follow the standard training protocol of a forward-facing scene in [11] to obtain the base model at the first frame. We start by a $256 \times 256 \times 128$ grid which is upsampled and sparsified for every 38.4K iterations until the model reaches the final resolution of $1408 \times 1156 \times 128$. The total training takes 128K iterations with a batch size of 5K rays. We adopt the RMSProp [19] optimizer with a decay parameter of $0.95$ for optimization. We adopt a slighter larger $TV$ penalty to mitigate foggy issues, where $\lambda_{TV}$ is set to $5 \times 10^{-4}$ for opacity and $5 \times 10^{-3}$ for color features.

**Per Frame Tuning.** The full-scale model and the pilot model are trained in parallel. For the experiments on the Meet Room dataset, we train the pilot model with 1000 iterations and then tune the full-scale model with 500 iterations. For the experiments on the N3DV dataset, we train the pilot model with 750 iterations and then tune the full model with 500 iterations. We use RMSprop to optimize both models with a batch size of 20k rays. To promote the pilot model to have a better convergence, we enlarge $\lambda_{TV}$ by 10 times when training the pilot model.

Table 1: **Comparison with related works on the N3DV and Meet Room datasets.** We report *per frame* PSNR, training and inference time and storage cost in average for a clear comparison. All the results are recorded with our 3090 GPU except the results of N3DV are referred to the numbers in the original paper. Compared to all the baselines, our method can achieve remarkable speedup on training time with competitive performance on the other metrics.

| Methods | N3DV | | | | Meet Room | | | |
| --- | --- | --- | --- | --- | --- | --- | --- | --- |
| | PSNR (dB) | Train (minutes) | Inference (seconds) | Storage (MB) | PSNR (dB) | Train (minutes) | Inference (seconds) | Storage (MB) |
| JaxNeRF*[13] | 28.53 | 485 | 67 | 14 | 27.11 | 473 | 40 | 14 |
| LLFF*[18] | 23.23 | 8 | **0.004** | 3192 | 22.88 | 3 | **0.003** | 1500 |
| N3DV [10] | **29.58** | 260 | >67 | **0.1** | - | - | - | - |
| Plenoxels*[11] | 28.68 | 23 | 0.12 | 4106 | **27.15** | 14 | 0.1 | 1015 |
| Ours | 28.26 | **0.25** | 0.12 | 17.7/31.4 | 26.72 | **0.17** | 0.1 | **5.7/9.0** |

\* Denote training from scratch per frame.

**Diff Based Compression.** We set the SH coefficient threshold $\epsilon$ to 1.5/27 and 1/27 for Meet Room Dataset and N3DV dataset respectively by the consideration of coefficient dimension (i.e., 27). We found using too small value would drastically increase saving size while the gain of rendering quality is marginal. In the experiments we use the about setting by default to achieve a proper trade-off between storage cost and rendering quality. In order to further reduce storage cost, we convert all saving SH coefficients and opacity to float16 and merge $\mathbf{M}_a^i$ and $\mathbf{M}_e^i$ to one single mask to save more space in the implementation. Moreover, as the mask is mostly comprised of consecutive binary values, it can be compressed with a very high compression ratio (over 99%) simply with a standard compression library (*e.g.* ZLIB[20]). We pack all tensors together and compress them with ZLIB to get the final on-disk storage cost for each frame.

**Pilot Model Guidance.** We first downsample the grid model and apply the standard training pipeline (narrow band and difference-based compression) for pilot model training. The induced model difference is used in two purposes, including filling the values of added and erased voxels back to the full-scale model and tuning the full-scale model with the guidance mask.

## 5.3 Experimental Comparison

We first quantitatively compare the method with some representative works including N3DV which achieves the state-of-the-art rendering quality. For the rest baselines, i.e., JaxNeRF [13], LLFF [18] and Plenoxels [11] which are originally designed and trained on static scenes, we train them in a per-frame independent manner, following the default setting as in their original papers. We use peak signal-to-noise ratio (PSNR), training and inference time and storage cost as metrics for a comprehensive evaluation, and report the mean scores per-frame. We refer to the scores reported in N3DV [10] for a fair comparison and estimate the rendering time of the method according to its network design. As it uses a larger MLP than the one in NeRF [7], its inference time is supposed to be longer than NeRF.

As shown in the Table 1, the implicit methods JaxNeRF and N3DV present fairly high computation cost, requiring over 8 GPU hours and 4 GPU hours for per-frame training, respectively. Our method can obtain remarkable advantage on training efficiency, which is three orders of magnitude faster than SOTA method N3DV. The inference efficiency also surpass them significantly. Unlike explicit baselines, our method can effectively decrease the need of storage.

Plenoxels and LLFF show the advantage on training and inference time benefiting from explicit representations, whereas they suffer from considerable storage cost (over several gigabytes per-frame). Our streaming modeling framework is capable of further enhancing training efficiency, i.e., obtaining ~100× acceleration over Plenoxels with comparable rendering quality, and ~20× acceleration over LLFF with obvious superiority in rendering quality. Moreover, our method can also surpass them obviously in storage cost. In a word, our method can outperform all the baseline methods substantially in terms of training efficiency, reaching a super fast speed (about 15 seconds and 10 seconds respectively) for per-frame optimization on the datasets. It is meanwhile able to attain competitive results on the metrics of rendering quality, inference time and storage.

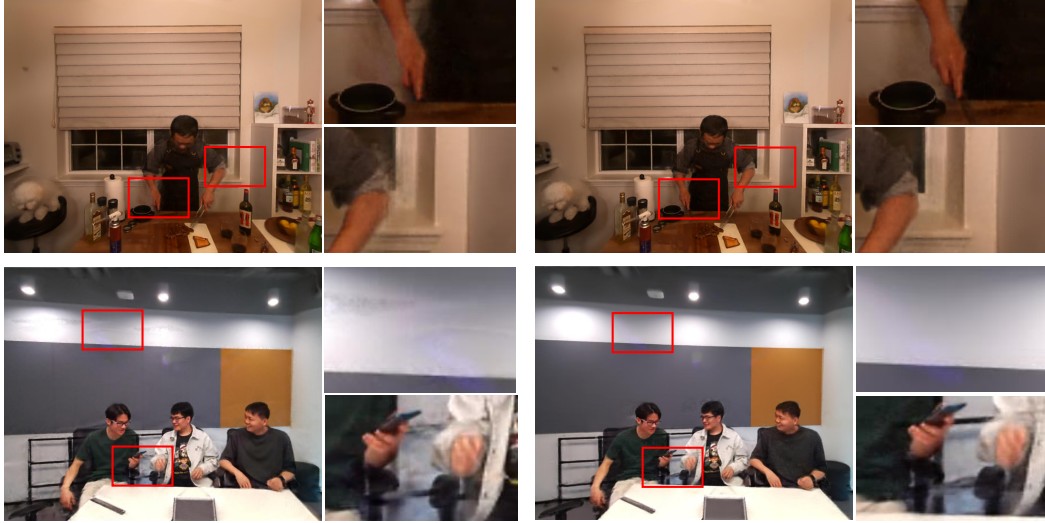

| w/o Pilot Model Guidance | w/ Pilot Model Guidance |

Figure 4: **Ablation study of pilot model guidance:** we compare **Left:** StreamRF trained without pilot model guidance and **Right:** with pilot model guidance in both N3DV dataset (top) and Meet Room dataset (bottom). These results reflect that pilot model can help reduce artifacts in both static background and dynamic foreground.

## 5.4 Ablation Studies

We conduct a series of ablation studies to further analyze and understand the method. More detailed experimental results and analysis can be referred to supplementary material.

**Narrow Band Finetune.** We validate the effect of using narrow band in Meet Room dataset, with the following two experimental configurations: (1) directly adopt the original sparse voxel grid, (2) switch the sparse grid of base model to its dense counterpart and tune on the dense grid. The curve of PSNR across frames and the visualization examples are shown in the Fig.2. We can observe that directly tuning a sparse grid is intractable to model difference caused by moving, resulting the rendering quality dropped immediately when motion happens out of active voxel grids, as shown in Fig.2 (a), though the right person only moves at a small distance. The rendering quality may be damaged obviously when there is a large movement, as shown in the Fig. 2 (b). As for training on a dense grid, it allows every voxel trainable. However, it still fails for modeling large movements with limited optimization steps. Compared to the sparse grid, a much large set of active voxels are tuned in the dense grid (5% vs. 100% of all the voxels) which might incur difficulty on convergence when training iterations are restricted. In contrast, finetuning with the narrow band strategy can effectively reach a stable and relatively high rendering quality across frames, even handling large movements. Moreover, our method can train around 40% faster and require 63% less memory consumption than tuning on the dense grid.

**Diff-based Compression.** We validate the performance of diff-based compression in Meet Room dataset, with the following configurations, i.e., training on the entire sequence, saving the models with and without diff-based compression. We then compare their performance on rendering quality (PSNR) and saving size (shown in Fig. 3 to investigate whether diff-based compression impairs the performance along with time steps (frames).

The quantitative result shows that our compression method is effective with negligible loss on rendering quality compared to un-compressed manner, and the performance can be highly preserved without degeneration to time. Compared to un-compressed version, using diff-based compression can effectively decrease the overall storage size from 1015 MB to an average of 5.7 MB for each frame, and the incremental storage for each frame is roughly stable along time steps. More details of model size reduction can be referred to supplementary material.

**Pilot Model Guidance.** We validate the strategy of using pilot model guidance visually in Fig. 4. For the base setting (right), we train the pilot model with 1,000 iterations and the full model with 500 iterations. For the ablation setting (left), we directly train the full model with 1,000 iterations to make per frame training time equivalent in both settings. As shown in Fig. 4, pilot model guidance help reduce various kinds of artifacts: (top) vanishing hands and blurry elbow; (bottom) sparkle noise in a white wall, the fuzzy area between two people, and achieve better visual quality than lengthening full-scale training with the same iterations (1500 iters with a ~30% increase on time cost). These results verified our intention that we could use an easy-to-optimized small model for benefiting full-scale model training. Pilot models can effectively indicate the voxels which need to be modified. Using this strategy can prevent unnecessary modification in the full-scale model and reduce the loss of details in traditional downsample-upsample process.

## 5.5 Limitation and Discussion

As shown in the Fig. 5, we can still observe artifacts like the loss of reconstructing high-frequency details and transparent/translucent objects in the rendering images which may "inherit" from using an explicit representation. It could be improved by extending the framework with explicit-implicit hybrid representations while remaining the benefit on training efficiency. Moreover, the rendering quality may also benefit from the use of some sophisticated designs, e.g., assigning adaptive bandwidth to different parts by employing high-level understanding on scenes via extra supervisions or leveraging multiple key-frames when applying it in real-word scenarios with extremely long sequences.

# 6 Related Work

**Static Novel View Synthesis.** Novel view synthesis learned from a set of input images is an important research topic in the field of computer vision and computer graphics. Earlier works like Lumigraph [21, 22] and Light-Field [23, 24] focus on interpolating the sampled light rays for realistic rendering. Some methods show that sampling density heavily depends on the underlying geometry of the captured scene [25, 26]. In this regard, many approaches are proposed to approximate [27, 28] or precise geometry model [29] to facilitate novel view rendering. Multi-plane image (MPI) [18, 30–32] as a representative direction is used to represent geometric information and enable realistic rendering of generic scenes.

More recently, NeRF [7] represents the radiance field with a multi-layer perceptron (MLP) network learned by differentiable volume rendering and realizes view interpolation with impressive quality. A series of methods are developed to improve it from different aspects. The works in [8, 33–35] focus on improving rendering quality by employing some tailored designs. Some methods [36, 37] are proposed to accelerate rendering speed by converting the large neural network to efficient representations with octree-based structure [38] or radiance maps [36] or a set of tiny MLPs [37]. Some approaches realize the boost of training efficiency by replacing the implicit formulation with explicit or hybrid representations [11, 39–42].

**Dynamic Novel View Synthesis.** Early efforts of generating 3D video can trace back to the 90s [4, 43] by computing explicit depth maps to realize novel view interpolation. By equipping with more sophisticated capturing hardware, the work in [1] can generate high-quality streamable free-viewpoint video. Matusik and Pfister [44] further integrate camera arrays and 3D displays to build an end-to-end 3D TV system.

More recent works learn a neural network to capture radiance fields of dynamic scenes. A family of methods [45–51] realize view synthesis by training on monocular videos captured by a moving camera, i.e., viewpoints change with respect to time steps. The works in [48–50] decouple the overall radiance field reconstruction to the learning of a static canonical template and a deformation field warping to the template. Park et al. [51] lifts canonical templates into hyper-spaces to induce flexibility on modeling variations in topology. Several methods model dynamics through the use of scene flows [45, 47] or depth prior [46], therefore they highly rely on the regularization of extra supervision (e.g., optical flow, depth, or foreground-background segmentation estimated by well-trained models). Broxton et al. [2] and Attal et al. [52] use multi-cameras stereo rig and extend MPI to MSI to support view interpolation with a larger field of view. Bansal et al. [53] use multiple mobile phones to capture a dynamic scene and compose the static and dynamic components with neural networks for 4D space-time visualization. The closest work to our setting is Neural 3D Video

(N3DV) [10], which captures dynamic scenes with synchronous video sequences from multiple cameras. It directly extends NeRF with an additional timecode input and accelerates computation with a hybrid importance sampling and hierarchical training. N3DV can produce strong visual results given enough optimization, while it is very time-consuming, requiring over per frame 5 GPU hours in training, making it impractical for real-world applications.

On the other hand, some approaches adopt the advances in neural rendering for the modeling of specific objects, e.g., human bodies [54, 55] by integrating with domain-specific priors, e.g., skeleton or SMPL mesh [56]. In contrast, our method imposes no prior in objects and is capable of representing more general scenes with complex deformation.

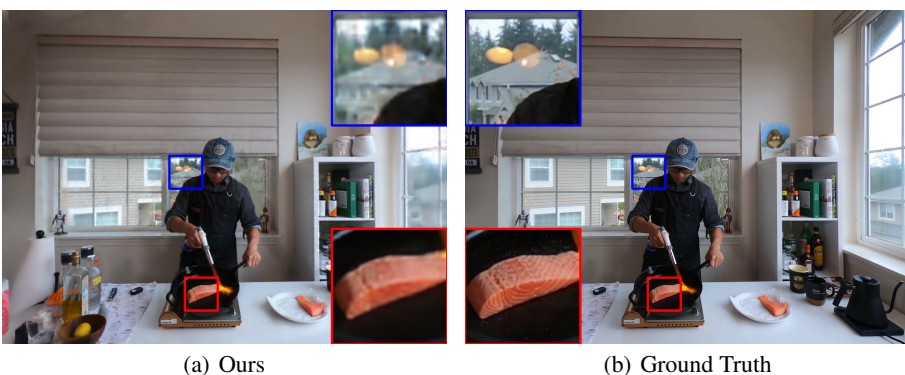

(a) Ours        (b) Ground Truth

Figure 5: **Failure Cases.** There are visual artifacts in reconstructing high-frequency details and transparent/translucent objects.

## 7 Conclusion

We propose a novel streaming radiance field method for effectively reconstructing and rendering dynamic scenes with explicit grid representations. The modelling of dynamic scenes is formalized as an incremental learning paradigm which allows the method for handling on-the-fly video sequences without need of recording scenes in advance. By virtue of narrow band tuning, our approach can achieve a super fast training convergence. The storage cost induced by the use of explicit grids can be obviously reduced by performing difference-base compression. We also present an efficient training mechanism with pilot model guidance to further improve model optimization. Experiments demonstrate that our approach is capable of training a high-performing model for dynamic scenes with the speed of 15s for tuning every frame, achieving ~1000× speedup over the state-of-the-art implicit dynamic methods. As a direction of future work, we expect to further accelerate the framework to support real-time training.

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
