# Streaming Radiance Fields for 3D Video Synthesis Supplemental Material

## 1 Meet Room dataset

We capture the Meet Room dataset by using 13 Azure Kinect DK [1] cameras. We use the view in centre for test and the rest views of 12 cameras for training. An example that captured by training views and the test view is shown in Figure 2 and Figure 3, respectively. All cameras are set to have identical exposure times. We use 3.5mm audio cables to form a daisy chain topology for shutter synchronization. We turn off depth cameras to downscale USB bandwidth. All the video sequences are captured with the $1280 \times 720$ resolution in 30 FPS. Figure 1 shows the capture system, and Figure 4 provides more examples in the Meet Room dataset.

Table 1: Model size reduction of each step in detail. All numbers are in MB.

|  | $M$ | $M_r$ | $V_r$ | $V_a$ | Total |
|---|---|---|---|---|---|
| baseline | 9.0 | 1.6 | 1466 | 4.9 | 1481.5 |
| +SH threshold | 9.0 | 1.6 | 8.0 | 4.9 | 23.5 |
| +float16 | 9.0 | 1.6 | 4.0 | 2.4 | 17.1 |
| +zlib | 0.4 | 0.1 | 3.0 | 2.2 | 5.7 |

Table 2: Performance and per frame training time compared with deformation-based methods.

| Methods | PSNR | Training time |
|---|---|---|
| Nerfies[2] | 26.96 | 13.2 min |
| Hypernerf[3] | 25.90 | 20.6 min |
| DynamicNeRF[4] | 27.87 | 16.8 min |
| D-NeRF[5] | 26.49 | 15.6 min |
| Ours | **28.26** | **0.25 min** |

## 2 Model Size Reduction

We show the details of model size reduction through each step in our diff-based compression in Table 1. We report the average size of all frames. A diff-based compression involves the following items: $M_e$, $M_a$ and $M_r$, the masks of erasing voxels, adding voxels and the remaining voxels while the features at the voxels may change. $V_a$ and $V_r$ denote the opacity and SH coefficients of adding and remaining voxels. In our implementation, $M_e$ and $M_a$ are directly inferred from the occupancy masks between adjacent frames by simple logical operation, hence we store the occupancy mask $M$ instead of $M_e$ and $M_a$.

## 3 More Comparison

Some methods [5, 4, 2, 3] are originally developed on modeling dynamic scene captured by monocular video. For a comprehensive study on our method, we extend them to multi-camera version and compare with them in the video sequences on the N3DV dataset [6]. Rendering quality (PSNR) and training cost (per-frame GPU hours) are listed in the Table 2. Our method can achieve significant improvement on training efficiency with better rendering quality.

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

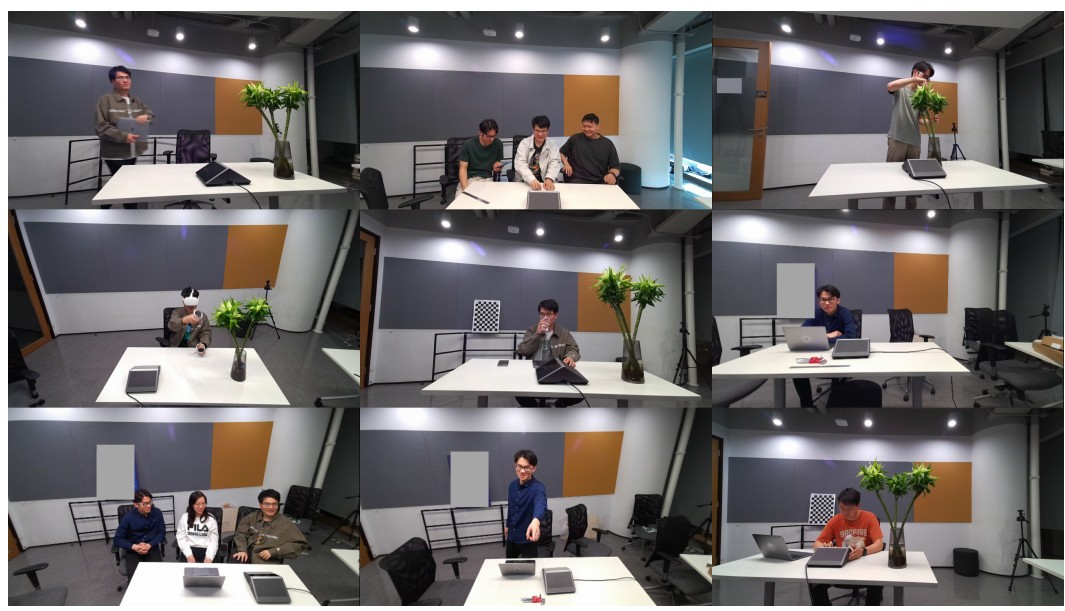

Figure 4: Examples of different scenes captured by our system.