# OpenReview forum: "Streaming Radiance Fields for 3D Video Synthesis"
_NeurIPS.cc/2022/Conference — NeurIPS 2022 Accept_

### Official Review · Reviewer_MbUF · 2022-07-10

**Rating:** 6
**Confidence:** 3
**Soundness:** 2 fair
**Presentation:** 3 good
**Contribution:** 3 good

**Summary:**

Neural radiance fields are not directly adaptible to dynamic scenes that have a changing representation that maps a 3D position to its captured image. The paper proposes a novel approach for capturing the radiance fields of video sequences. The radiance field of the base model (first frame) is learned explicitly using a voxel-grid parameterization. Consequently, only the frame-to-frame differences in the grid representation are learned. The difference between frame-grids is identified using a narrow band focused around the contour of the dynamic segments of the scene. Furthermore, a compression strategy is proposed to handle the explosion in size caused by the use of explicit voxel grids. Finally a multi-scale learning paradigm is introduced for improved optimization during learning the proposed model.


**Questions:**

Apart from the computational benefit, what would be a theoretical motivation to compare the pilot model base training as opposed to training at full scale.

**Limitations:**

No, the authors have not adequately discussed limitations within the paper.

**Strengths And Weaknesses:**

Strengths:
The paper is well written and the exposition is clear.
The narrow-band tuning strategy proposed in the paper is intuitive and properly motivated within the context of local changes in adjacent dynamic video frames.
The compression strategy based on the mask difference is also an interesting contribution of the paper and a useful construct for dynamic video representation.

Weaknesses:
The pilot model guidance framework, on the other hand, requires better exposition and motivation. It is clear that multi-scale parameterization has a computational impact on problems that use dense grid representations. However, the qualitative impact may be larger, and possibly even negative. For example, there needs to be more discussion on the impact of coupling the pilot model approach with the narrow-band tuning. Given the dynamism of the scene and the arbitrary position of the target viewpoint, it is not difficult to imagine that full scale training is qualitatively superior to a multi-scale approach that might suffer from loss of detail.

---

> ### Author Response · Authors · 2022-08-02
> **Reply to Reviewer MbUF**
>
> We thank the reviewer for the valuable suggestions and address the reviewer's concerns as follows.
>
> **Q1 The pilot model guidance framework requires better exposition and motivation.**
>
> We apply a streaming framework. For each incoming frame, we first downsample the grid model and apply the standard training pipeline (narrow band + difference-based compression). The induced model difference is used in two purposes: 1) the values of added and erased voxels are filled back to the full-scale model. 2) the guidance mask is used to regularize the gradient update when tuning the full-scale model. We have added more descriptions in the section 4 and 5.2 (implementation details).
>
> The design of pilot model guidance comes from the insight that training at a small size converges much faster and the induced difference of pilot model indicates the voxels which need to be modified. Using this strategy can prevent unnecessary modification in the full-scale model and avoid the loss of high-frequency details during downsample-upsample process. The qualitative results in the Fig. 4 show the comparison via such a multi-scale training strategy (1000 iters for training pilot model and another 500 iters for training the full-scale model) versus directly full-scale training (1000 iters) with a similar time cost. The rendering results of full-scale setting exist noticeable artifacts in both static background and dynamic foreground. Using pilot model guidance can help reduce these artifacts and achieve better visual quality than lengthening full-scale training with the same iterations (1500 iters with a ~30\% increase on time cost).

---

### Official Review · Reviewer_4vFM · 2022-07-11

**Rating:** 6
**Confidence:** 4
**Soundness:** 3 good
**Presentation:** 4 excellent
**Contribution:** 4 excellent

**Summary:**

This paper presents a grid-based method to reconstruct a streaming radiance field for dynamic scenes. More precisely, this paper formulates the dynamic scene modeling problem in an incremental learning paradigm where per-frame model difference is trained to complement the adaption of a base model (the first frame). With the above design, the proposed method can efficiently handle on-the-fly video sequences without significant storage overhead. The optimization can be further accelerated via a curriculum learning strategy. Experiments on challenging video sequences demonstrate that the proposed method achieves 1500x speedup over state-of-the-art implicit models.

**Questions:**

(1)	Will the first frame always be optimal as the base model? Have you compared it with using different frames (middle, last) and learning the relative difference? Will this method be used with multiple base frames? For example, one alternative is to learn full grid models at several “key frames” and all other frames can be seen as interpolations from these key frames. I expect this would improve some performance especially when the objects have large movements.
(2)	How are the narrow bands hyper-parameters (rou_d and rou_e) selected for different videos? I assume the narrow bands correspond to the velocity of the object in the video, and inaccurately choosing a too small band will cause errors. Is it possible to adaptively assign the narrow bands for different parts?
(3)	Can a similar approach be used in other hybrid multi-resolution representations such as Instant-NGP? What will be the major difficulties?


**Limitations:**

This paper did not discuss any potential negative societal impact. Since the proposed method can be used to reconstructing/learning any Internet videos, it may be better to add the discussion on the social impact.

**Strengths And Weaknesses:**

Strengths:
This paper proposes a novel approach of using explicit grids for reconstructing streaming radiance fields, which significantly improves over exiting implicit methods, enabling learning on-the-fly videos in a streaming manner. Also, by only storing the difference for each frame, the complete model can be efficiently compressed to reduce the memory costs brought by explicit grid models.
The pilot model further improves the training efficiency by first learning a low-resolution pilot model, and then using the guidance from the pilot model to learn in full resolution.
Therefore, the proposed methods achieve a good balance between explicit and implicit methods in terms of learning efficiency and storage.

Weaknesses:
Although the output results show improvement over the baselines both qualitatively and quantitively, we can still see obvious artifacts in the supplementary video. There are scenes the people get transparent or overlapped in the space. The authors need to add more discussions and explanations on the artifacts and check if these phenomena are introduced by imperfect learning in model differences and error accumulation.
This paper did not have a comparison with implicit models which learns deformation between frames (e.g., NeRFies, D-NeRF, DynamicNeRF, HyperNeRF, Neural Actor, AnimatableNeRF, etc). All these methods also use the inductive bias of local continuity and use a neural network to learn the deformation field for different frames. These methods may have longer convergence (due to implicit based), however, may achieve better quality and fewer artifacts for moving objects.

Missing references:
1. Hypernerf: A higher-dimensional representation for topologically varying neural radiance fields
2. Nerfies: Deformable neural radiance fields
3. Non-Rigid Neural Radiance Fields: Reconstruction and Novel View Synthesis of a Dynamic Scene From Monocular Video
4. D-NeRF: Neural Radiance Fields for Dynamic Scenes
5. Neural Sparse Voxel Fields
6. Neural Actor: Neural Free-view Synthesis of Human Actors with Pose Control
7. Animatable Neural Radiance Fields for Modeling Dynamic Human Bodies

---

> ### Author Response · Authors · 2022-08-02
> **Reply to Reviewer 4vFM**
>
> We thank the reviewer for the valuable suggestions and address the reviewer's concerns as follows.
>
> **Q1 see obvious artifacts in the supplementary video. There are scenes the people get transparent or overlapped in the space. The authors need to add more discussions and explanations on the artifacts.**
>
> We observed similar artifacts like translucent foreground and texture sticking in the explicit counterpart Plenoxels which is well-trained for each single frame, indicating that the issues may "inherit" from Plenoxels. The capture viewpoints in Meet Room are sparse (13 views including one view for testing)  and arranged in a wider baseline. Less textured background and more rapid motion with topological changes also pose challenges for achieving high-fidelity rendering. We have added some discussion in the revised conclusion section.
>
> **Q2 This paper did not have a comparison with implicit models which learns deformation between frames.**
>
> We didn't include the comparison with these methods mainly due to setting difference. Our method focuses on  the scenes captured by multiple fixed cameras with known poses (Line 16). To our best knowledge, N3DV achieves SOTA performance with the same setting to ours, hence we treat it as the counterpart in implicit methods. Nerfies, HyperNeRF, D-NeRF, NR-NeRF and DynamicNeRF realize view synthesis with the setting of monocular moving camera, i.e., viewpoints change w.r.t time steps.
>
> To address the reviewer's concern, we extend the published code of Nerfies, HyperNerf, D-NeRF and DynamicNeRF to multi-view version and compare with them in the video sequences released in N3DV for a fair comparison. The performance (PSNR) and training cost (per-frame GPU hours) are listed as follows. Compared to their default setting, our method can achieve two orders of magnitude improvement on training efficiency with better rendering quality. The rendering quality of these methods may benefit from using larger models with lengthened training schedule and hyper-parameter tuning, which would be beyond the scope of this work.
>
> | Methods     | PSNR  | perframe training time |
> | ----------- | :---: | :--------------------: |
> | Nerfies     | 26.96 | 13.2 min               |
> | Hypernerf   | 25.90 | 20.6 min               |
> | DynamicNeRF | 27.87 | 16.8 min               |
> | D-NeRF      | 26.49 | 15.6 min               |
> | N3DV        | 29.58 | 260 min                |
> | Ours        | 28.16 | 0.17 min               |
>
>
> **Q3 missing reference**
>
> We have cited and discussed them in the revision of related work.
>
> **Q4 Will the first frame always be optimal as the base model? Have you compared it with using different frames (middle, last), Will this method be used with multiple base frames?**
>
> As our method targets on a streaming scenario, training from the first frame is a necessary strategy. We also tested on training from either middle or last frames and observed negligible difference in terms of PSNR and visual quality. We agree with the reviewer that using multiple key-frames might further improve the ability when dealing with extremely long video sequences with complicated scenarios (e.g., dramatically changing foreground or background) in real-world applications which we opt for leaving as a future work. We have added some discussion in the revision.
>
> **Q5 How are the narrow bands hyper-parameters (rou\_d and rou\_e) selected for different videos?**
>
> We have tested the setting of 1, 2, and 3 to both rou\_d and rou\_e, and observed minimal quantitative difference (26.69, 26.79 and 26.74). We can observe clear artifacts when large and rapid motions happen on the setting of 1. Using a wider band would also reduce efficiency.  We choose 2 by default for all the experiments.
>
> **Q6 Is it possible to adaptively assign the narrow bands for different parts?**
>
> Thanks for the suggestion. It would be a valuable direction to further boost StreamRF but assigning adaptive bandwidth to different parts may build upon high-level understanding on scenes or extra supervisions (e.g., optical flows). We have discussed it in the revision.
>
>
> **Q7  Can a similar approach be used in other hybrid multi-resolution representations such as Instant-NGP? What will be the major difficulties?**
>
> If adapting the approach to Instant-NGP, the main difficulty is to extract narrow band from its multi-resolution representations which are embedded into a hashing table in a random manner.

---

### Official Review · Reviewer_nDG4 · 2022-07-12

**Rating:** 5
**Confidence:** 4
**Soundness:** 3 good
**Presentation:** 2 fair
**Contribution:** 3 good

**Summary:**

The presented work describes the extension of Plenoxels [1] to temporal sequences. Compared to fitting one Plenoxel grid per frame, the proposed method improves memory and training time requirements significantly, by only learning to estimate the difference between adjacent frames.

The paper claims the following technical contributions
 - A time and memory-efficient extension of Plenoxels to dynamic scenes by fitting frame-to-frame differences.
 - Only training in a "narrow band" where scene motion is expected
 - Multi-scale training based on a "pilot model"
 - Compression of frame-to-frame differences
 - A new multi-view dataset captured with a rig of 13 Azure Kinect cameras

[1] Yu et al. Radiance Fields without Neural Networks. 2022


**Questions:**

 - In Eq. 2, should it be Y(d)?


**Limitations:**

Limitations are adequately discussed.

**Strengths And Weaknesses:**

The paper addresses an important problem, the long training times of state of the art neural 3D video reconstruction [2], whose authors report 1,300 GPU hours of training time for a 10 second clip. The authors show how to adapt Plenoxels to video reconstruction and demonstrate training and inference times that are significantly faster than [2], and still achieve reasonable reconstruction quality.

The quantitative results reported in Table 1 are convincing. Visually, the results on the dataset of [2] are good, although they don't achieve the quality of [2]. The results on the proposed "Meet Room" dataset exhibit more artifacts (e.g. translucent foreground and streak aftifacts along edges), which might be caused by a wider camera baseline, a less textured background, and might also be the results of some of the performance optimizations (e.g. narrow band finetuning) that the authors employ.

Overall, the paper is well structured and the discussion of related work is comprehensive. However, the discussion of the technical contributions is unfortunately lacking clarity:

## Pilot model

The authors propose to fine-tune step $N+1$ first on a lower-resolution model, before initializing the higher resolution model from the pilot model. This seems very similar to the coarse-to-fine optimization described by Plenoxels. Could the authors comment on why this should be considered a novel contribution?

## Narrow Band Finetuning

It is not clear to me why the "dense grid" column in Figure 2 fails the way it does. The failure of the "sparse grid" column is plausibile, since it is using the occupancy map of the original frame. I would expect training using the dense grid to work better (although) than selecting the narrow band. Why does using the narrow band increase quality?

The paper would benefit from a figure analyzing the performance benefit of narrow-band finetuning and the Pilot model.

## Difference-based compression

 - Table 1 should include the model size for all techniques.
 - The compression steps in 5.2 (SH thresholding, float16 coefficients, ZLIB) are sound contributions, but are not sufficiently detailed. The paper should detail the model size reduction of each step.
  - How big is the model size of the sparse fine-tuned voxels before compression? Line 155 suggests the finetune output has a size of 5.4 MB, but this equals the size of the fully compressed model reported elsewhere.
  - How are the voxels selected for narrow band tuning different from the voxels that are included in difference-based compression? The paper would benefit from a visualization of both selection criteria, and a measurement of any extra compression gained by Eq. 6.



[2] Li et al. Neural 3D Video Synthesis from Multi-view Video. 2021

---

> ### Author Response · Authors · 2022-08-02
> **Reply to Reviewer nDG4**
>
> We thank the reviewer for the valuable suggestions and address the reviewer's concerns as follows.
>
> **Q1  why pilot model should be considered a novel contribution.**
>
> We apply a streaming framework. For each incoming frame, we first downsample the grid model and apply the standard pipeline (narrow band + difference-based compression) for training the pilot model. The induced model difference is used in two purposes: 1) the guidance mask is used to regularize the gradient update of full-scale model, where the voxels outside the mask are frozen without optimization. 2) the values of added and erased voxels (rather than the entire pilot model) are filled back to the full-scale grid model.
>
> The design of pilot model guidance comes from the insight that training at a small size converges much faster, hence the induced difference of pilot model indicates the voxels which need to be modified. Using this strategy can prevent unnecessary modification in the full-scale model and avoid the loss of high-frequency details during downsample-upsample process. The technical implementation and the motivation are both distinct from coarse-to-fine (C2F) strategy in Plenoxels.  If we follow the C2F strategy by directly upsampling the small-scale model for tuning the full-scale one, the quality is inferior with 0.3 dB drop in PSNR and noticeable artifacts. We have added more descriptions in the section 4 and 5.2 (implementation details).
>
> **Q2  Why does using the narrow band increase quality?**
>
> The effect of using narrow band does not expect to increase the upper bound of quality (compared to per-frame training baseline). Instead it facilitates convergence and is able to achieve more convincing results with limited training time. In figure 2, we compare training with "dense grid", "sparse grid" and "narrow band finetune" under the same setting of 1K iterations. The results show that training with "narrow band finetune" can achieve a better result given the same training budget. We have observed that compared to "narrow band finetune", training with "dense grid" needs 10X training iterations to achieve the same quality where each step typically costs 2X computational time, therefore it needs to cost 20X time. We will make this part more clear in the revision.
>
>
> **Q3 Table 1 should include the model size for all techniques.**
>
> The model sizes of different methods are listed as follows. Note that our method only saves a full-scale sparse grid model for the first frame and store model difference for the following frames. The size of per-frame model difference is ~5.4 MB in average and the size is ~8.8 MB in average when counting the full-scale grid model.
>
> | methods   | per frame mode size (MB) |
> | --------- | :----------------------: |
> |  JaxNeRF  | 14                       |
> | LLFF      | 1500                     |
> | N3DV      | 0.1                      |
> | Plenoxels | 1015                     |
> | Ours      | 5.4 / 8.8                |
>
> **Q4 The paper should detail the model size reduction of each step.**
>
> The model size of each step is listed in the table. We report the average size of all frames. A diff-based compression involves the following items: $M_e$, $M_a$ and $M_r$, the masks of erasing voxels, adding voxels and the remaining voxels while the features at the voxels may change. $V_a$ and $V_r$ denote the opacity and SH coefficients of adding and remaining voxels. In our implementation, $M_e$ and $M_a$ are directly inferred from the occupancy masks between adjacent frames by simple logical operation, hence we store the occupancy mask $M$ instead of $M_e$ and $M_a$ (Line 218).
>
> |               | baseline    | +SH thresholding | +float16 | +zlib  |
> | ------------- | ----------- | ---------------- | -------- | ------ |
> | $M$            | 9.0 MB      | 9.0 MB           | 9.0 MB   | 0.4 MB |
> | $M_r$           | 1.6 MB      | 1.6 MB           | 1.6 MB   | 0.1 MB |
> | $V_r$           | 1466 MB     | 10.5 MB          | 5.2 MB   | 3.7 MB |
> | $V_a$          | 2.5 MB      | 2.5 MB           | 2.2 MB   | 1.2 MB |
> | Total         | 1479.1 MB   | 23.6 MB          | 18.2 MB  | 5.4 MB |
>
> **Q5 How big is the model size of the sparse fine-tuned voxels before compression?  Line 155 suggests the finetune output has a size of 5.4 MB.**
>
> We have fixed the typo in the revision. The size of the sparse grid model is 1015 MB.
>
> **Q6 How are the voxels selected for narrow band tuning different from the voxels that are included in difference-based compression?**
>
> The selected voxels in the compression actually forms a sub-set of the ones in the narrow band tuning. If we adapt the same strategy (i.e., float16 conversion and zlib compression) to directly saving all the voxels within the narrow band, the per-frame model size would be ~426 MB in average.  We have add a figure in appendix to better illustrate the mechanism of difference-based compression.
>
> **Q7 In Eq. 2, should it be Y(d)?**
>
> We have fixed the typo Y(d) in the revision.

---

### Meta-Review · Area_Chair_Qo8W · 2022-08-30

**Recommendation:** Accept
**Confidence:** Certain

**Metareview:**

All reviewers agree that the paper should be accepted, despite some flaws that can be addressed in future work

**Award:**

No

---

### Decision · Program_Chairs · 2022-09-14

Accept